# How coping styles moderate the relationship between diabetes distress and self-care in adults with diabetes in Appalachian Ohio: A cross-sectional survey study

Elizabeth A. Beverly[1,2][*], Samuel Miller[3], Lauren Schmidtgesling[2], Kacey Whistler[4]

1 Department of Primary Care, Ohio University Heritage College of Osteopathic Medicine, Athens, Ohio, United States of America, 2 The Diabetes Institute, Ohio University, Athens, Ohio, United States of America, 3 Department of Medicine, Ohio University Heritage College of Osteopathic Medicine, Athens, Ohio, United States of America, 4 The Graduate College, Ohio University, Athens, Ohio, United States of America

☯ These authors contributed equally to this work.

* beverle1@ohio.edu

## Abstract

### Background

Diabetes distress is highly prevalent among adults with type 1 and type 2 diabetes, and is associated with fewer self-care behaviors, higher A1C values, increased complications, and lower quality of life. The purpose of this study was to confirm the moderating role of problem-based and emotion-based coping styles in the relationship between diabetes distress and self-care among adults living in Appalachian Ohio. We hypothesized that problem-based coping buffered, while emotion-based coping exacerbated the negative impact of diabetes distress on self-care.

### Methods

Participants completed a series of psychosocial questionnaires, including the Diabetes Distress Scale, Coping Styles, and Self-Care Inventory-Revised. We conducted hierarchical multiple regression models using SPSS version 29.0.

### Results

A total of 256 adults (mean age = 42.6 ± 18.7 years, 62.1% women, 61.3% type 2 diabetes, A1C = 7.5 ± 1.5%, mean duration = 12.1 ± 8.9 years) participated. In the type 1 diabetes model, the two-way interactions between problem-based coping and diabetes distress (B = 0.611, p = 0.011) and emotion-based coping and diabetes distress (B = 0.379, p = 0.017) were significant with diabetes self-care (R2 Δ = 0.105, p = 0.002); the effect size was F² = 0.290, indicating a medium effect. In the type 2 diabetes model, the two-way interaction between problem-based coping and diabetes distress

**Data availability statement:** All relevant data for this study are publicly available from the Mendeley Data repository (https://doi.org/10.17632/rrx6stxcfk.1).

**Funding:** American Osteopathic Association Grant: Osteopathic Approach to Diabetes: Pathophysiology, Treatment, Outcomes and Complication Grant (No. 1291708718).

**Competing interests:** NO authors have competing interests.

(B = 0.388, p = 0.012) was significant with diabetes self-care (R2Δ=0.044, p = 0.018); the effect size was $F^2$ = 0.198, indicating a medium effect.

## Conclusions

Our findings suggest that higher levels of problem-based coping buffered the negative effect of diabetes distress on self-care behaviors in both type 1 and type 2 diabetes. In contrast, emotion-based coping amplified the negative effect of diabetes distress on self-care behaviors, but only in participants with type 1 diabetes. These findings align with previous research and intervention studies focused on coping strategies and diabetes distress.

## Introduction

Diabetes distress is the most common psychosocial concern experienced by people with diabetes. It refers to the emotional burden of managing a complex, chronic condition over time [1]. Diabetes distress encompasses frustrations with self-care, fears about developing complications, and concerns with access to care [2]. It is a natural response to the challenges of managing diabetes, contributing to its high prevalence. In adults with type 2 diabetes, the prevalence of diabetes distress ranges from 18% to 45% [3], with newer data suggesting rates may be as high as 60% [4]. Among adults with type 1 diabetes, the prevalence of diabetes distress ranges from 22% to 42%, with newer data indicating rates may be as high as 70% [5–7]. In rural Appalachian Ohio, the prevalence of diabetes is 19.9% [8], nearly double the United States national average, and high diabetes distress rates are 31% in adults with type 1 diabetes and 28% in adults with type 2 diabetes [9]. Importantly, diabetes distress is not a psychiatric disorder, although it frequently co-occurs with depressive symptoms [10] and anxiety symptoms [11].

Clinically, diabetes distress has unique relationships with glycemic and other outcomes. In adults with type 2 diabetes, diabetes distress is associated with higher Hemoglobin A1c (A1C) levels both cross-sectionally and longitudinally [12,13]. It is also associated with hypertension [14], higher low-density lipoprotein cholesterol (LDL-C) levels [15], microvascular complications [14–16], and macrovascular complications [14,17]. Furthermore, it is associated with performing fewer self-care behaviors [15,18,19] and lower quality of life [9,20]. These associations may explain the observed association between diabetes distress and increased risk for all-cause mortality in type 2 diabetes [16]. In adults with type 1 diabetes, diabetes distress is also associated with higher A1C levels both cross-sectionally and longitudinally [6,21,22]. Similarly, diabetes distress is associated with higher total cholesterol levels, [22] microvascular complications, [6] and macrovascular complications [23], performing fewer self-care behaviors [23,24] and lower quality of life in adults with type 1 diabetes [25–27]. Given the substantial clinical implications of diabetes distress, strategies to reduce it are critical. This need is particularly important in rural regions such as Appalachian Ohio, where healthcare access is limited. Compared with national

averages, Appalachian Ohio has 12% fewer primary care physicians, 28% fewer physician specialists, and 35% fewer mental health providers per 100,000 people, and the diabetes mortality rate is 28.4% higher [28,29].

Coping styles may be a target for interventions given their association with diabetes distress in both adults with type 1 and type 2 diabetes [30–33]. Coping refers to the strategies individuals use to manage threats posed by a stressor [34]. People typically use a combination of problem-based and emotion-based coping styles [34]. Problem-based coping involves concrete actions to address the problem or stressor, such as planning, seeking information, and/or directly confronting the problem, and emotion-based coping attempts to address the emotional response to the stressor [34]. Prior research has shown that problem-based coping is associated with lower A1C levels [35] and lower diabetes distress scores [32], whereas emotion-based coping is associated with fewer self-care behaviors [36], anxiety symptoms [31], depressive symptoms [31,33], and higher diabetes distress scores [32,33]. Building on this research, problem-based coping may buffer diabetes distress while emotion-based coping may exacerbate diabetes distress.

Thus, the purpose of this study was to confirm the moderating role of problem-based and emotion-based coping in the relationship between diabetes distress and self-care in adults with type 1 and type 2 diabetes in a rural population from Appalachian Ohio. We hypothesized that problem-based coping would positively buffer the relationship between diabetes distress and self-care in adults with type 1 and type 2 diabetes. Specifically, as problem-based coping increased, we expected the negative effects of diabetes distress on self-care to decrease. We also hypothesized that emotion-based coping would exacerbate the relationship between diabetes distress and self-care, such that higher levels of emotion-based coping would amplify the negative effect of diabetes distress on self-care in adults with type 1 and type 2 diabetes.

## Methods

In this descriptive, cross-sectional survey study, we investigated how coping styles moderate the relationship between diabetes distress and self-care in adults with type 1 and type 2 diabetes from rural Appalachian Ohio. Specifically, we examined the interaction effects of problem-based coping and emotion-based coping on diabetes self-care behaviors. The Ohio University Office of Research Compliance approved the protocol (17X234) on August 18, 2017, and all recruitment procedures and materials.

### Participants

We recruited adults aged 18 years and older who could read and speak English, had been formally diagnosed with type 1 and type 2 diabetes by a health care professional, and were currently residing in southeastern Ohio (i.e., Athens, Hocking, Meigs, Morgan, Perry, Vinton, and Washington counties). Exclusion criteria included individuals whose diabetes diagnosis could not be definitively classified as type 1 and type 2 diabetes, under 18 years of age who could not read or speak English, and those not currently residing in the seven counties of southeastern Ohio. These seven counties are part of Appalachia, a 205,000-square-mile region that encompasses 420 counties in 13 states from New York to Mississippi. The Appalachian Region is 42% rural compared to 20% of the US as a whole [37]. In southeastern Ohio, 17.0% of the population lives below the poverty line as compared to 11.1% of the country [38], and the poverty rates range from 16.4% to 25.0% [39]. Moreover, these seven counties are designated as health professional shortage areas for primary care, dental care, and mental health providers [29]. People living in these counties are more likely to be unemployed, have lower educational achievement, and limited access to transportation [40]. Related to diabetes, people are more likely to have a delayed diabetes diagnosis, macrovascular and microvascular complications, lower limb amputations, and depression [28,41].

We recruited participants via email outreach, flyers, and advertisements. Recruitment emails were distributed to all employees at the University, which is the largest employer in the seven-county region, and to individuals listed in the University Diabetes Institute registry who had previously given permission to be contacted for future research opportunities.

In addition, study flyers were posted in public locations, including grocery stores, libraries, primary care offices, and the seven county health departments. Finally, written advertisements were printed in the local newspaper and broadcast on the local public radio station to increase recruitment reach and participation across the seven-county region. Eligible participants were provided an electronic anonymous survey or, upon request, mailed a survey packet. The study opened on December 14, 2017, and concluded on August 18, 2019. Participation was entirely voluntary.

**Measures**

Participants filled out a brief demographic form that collected self-reported sociodemographic and health information, including age, gender, race, ethnicity, education level, health insurance coverage, type of diabetes, Hemoglobin $A_{1C}$ (A1C) level, body mass index (BMI), diabetes duration, and diabetes medications.

Participants also completed the following measures

Type 1 Diabetes Distress Scale [42]: This 28-item scale assesses seven sources of diabetes distress commonly experienced by adults with type 1 diabetes. These domains include powerlessness, negative social perceptions, physician distress, friend/family distress, hypoglycemia distress, management distress, and eating distress. Respondents rate each item on a 6-point scale, ranging from "not a problem" to "a very serious problem." Items were averaged, and the cut-points were little or no distress (1.0–1.4), mild distress (1.5–1.9), moderate distress (2.0–2.9), and high distress (≥3.0) [42]. The scale has demonstrated good reliability (α = 0.91, sub-scale range α = 0.76–0.88), and strong test-retest reliability (r = 0.74) [42].

Type 2 Diabetes Distress Scale [43]: This 17-item scale assesses four sources of diabetes distress common commonly experienced by adults with type 2 diabetes. These domains include emotional burden, physician-related distress, regimen-related distress, and interpersonal distress. Respondents rate each item on a 6-point scale, ranging from "not a problem" to "a very serious problem." Items were averaged, and the cut-points were little or no distress (1.0–1.4), mild distress (1.5–1.9), moderate distress (2.0–2.9), and high distress (≥3.0) [44]. The scale has demonstrated good reliability (α = 0.93, subscales range α = 0.88–0.90) [43].

Patient Health Questionnaire-9 (PHQ-9) [45]: This is a 9-item instrument that assesses the nine diagnostic criteria for major depressive disorder as outlined in the *Diagnostic and Statistical Manual of Mental Disorders-5th Edition* (DSM-5). These criteria include 1) anhedonia, 2) depressed mood, 3) sleep disturbances, 4) fatigue, 5) change in appetite, 6) feelings of guilt, self-blame, or worthlessness, 7) difficulty concentrating, 8) psychomotor agitation or retardation, and 9) suicidal ideation. [45] Respondents indicated how often they have experienced these symptoms in the past two weeks using a 4-point scale, including "Not at all, Several days, More than half the days, Nearly every day." Total scores were summed with scores ranging from 0 to 27.

Self-care Inventory-Revised [46]: This 15-item scale assesses the frequency of self-care behaviors. Respondents rated the frequency of their self-care behaviors on a 5-point scale, with options ranging from "never" and "rarely" to "sometimes," "usually," and "always." The self-care inventory-revised total score was calculated as the mean, with higher scores reflecting greater perceived engagement in self-care behaviors.

Coping Styles [47]: This 15-item questionnaire originally developed by Wilson et al. [48] and adapted by Peyrot and colleagues [47,49,50] identified two higher order dimensions of coping: problem-based coping and emotion-based coping. Problem-based coping strategies included stoicism and pragmatism, demonstrated through statements about managing one's emotions and problem-solving techniques to reduce frustration. Emotion-based coping strategies included behavior such as anger, impatience, and anxiety, characterized by expressions of anger, impulsive actions, anxious responses (nervousness, difficulty relaxing) and avoidant behaviors (giving up or avoiding tasks). Respondents rated each item on a 4-point scale, ranging from "not at all like me" to "very much like me." Subscale scores for problem-based coping and emotion-based coping were calculated as means, with higher scores indicating greater endorsement of that coping style. This measure has been validated in populations with diabetes. [47]

Confidence in Diabetes Self-Care Scale [51] This 20-item scale assesses self-efficacy in diabetes self-care, which is the confidence individuals have in their ability to perform self-care behaviors. Respondents rated each item on a 5-point scale, ranging from "No, I am sure I cannot" to "Yes, I am sure I can." The scale has demonstrated high internal reliability ($\alpha = 0.90$). [51] Total scores were summed ranging from 0 to 100, with higher scores indicating greater perceived self-efficacy to perform self-care. Self-efficacy is an important component of activation/empowerment. [52–55]

Medical Outcomes Study Social Support Survey [56] This 19-item survey assesses social support using a 5-point Likert scale, producing a total social support score along with four subscales: 1) emotional/informational support (e.g., empathetic understanding, encouragement of expression, advice, guidance), 2) tangible support (e.g., the provision of material aid or behavioral assistance), 3) affectionate support (e.g., expressions of love and affection, 4) positive interaction social support (e.g., availability of others to do fun things with you). All items are worded positively and total scores were calculated as means, with higher scores indicating more perceived social support. The survey has demonstrated high internal reliability ($\alpha = 0.91$) [56].

## Data collection

Participants completed the survey either through the online platform Qualtrics (Provo, Utah) or mailed survey packets. Electronic informed consent was obtained by having participants select a radio button indicating "Yes, I consent to participate in this study. I may withdraw my participation at any time." Those who declined clicked a button stating, "I decline to participate." To avoid coercion, the online screen to the survey and the informed consent document both specified the voluntary nature of participation. For the mailed survey packets, participants provided written informed consent prior to participating in the survey study. Researchers were not present when potential participants decided whether to participate or decline. All participants provided electronic or written informed consent. Completion of the survey took approximately 30–45 minutes. Participants received a $15.00 gift card as compensation.

## Diabetes diagnosis verification

To enhance the accuracy of self-reported diabetes diagnoses, we conducted a multi-step data verification process. This process involved sequential checks for medication congruence, alignment of diabetes distress scales with type of diabetes, consistency between age and diabetes duration, plausibility of A1C values, and identification of duplicate entries prior to final analyses (see S1 Fig). At the end of our multi-step process, 157 cases were excluded from the initial sample (n = 413). Prior to starting this process, we excluded 88 cases because they consented but did not complete any demographic or survey questions. In the first step, nine type 1 diabetes cases were removed due to medication mismatch and 16 type 1 diabetes cases were removed due to missing medication data (n = 105); no type 2 diabetes cases were removed (n = 195). In the second step, one type 1 diabetes case completed the Type 2 Diabetes Distress Scale and was excluded (n = 104); one type 2 diabetes case completed the Type 1 Diabetes Distress Scale and was excluded (n = 194). In step 3, three type 1 diabetes cases listed a duration longer than their age and were excluded (n = 101); no type 2 diabetes case listed a duration longer than their age (n = 194). In step 4, no type 1 diabetes case self-reported A1Cs that were implausible, but two cases were missing A1C data and were excluded (n = 99). One type 2 diabetes case reported an implausible A1C value and was excluded, and 27 cases were missing A1C data and were excluded (n = 166). In step 5, no type 1 diabetes entries were duplicated (n = 99); nine type 2 diabetes entries appeared to be duplicated and were removed from the dataset (n = 157). The removal of these cases also eliminated nearly all instances of missing data, thereby enhancing the overall integrity of the dataset.

## Data analysis

We used descriptive statistics to assess sociodemographic and health characteristics. Data were assessed for normality and skewness using histograms and the Kolmogorov–Smirnov test. The examination of histograms and skewness

statistics indicated that A1C data were right-skewed. Therefore, A1C data were $\log_{10}$-transformed to correct for right skewness to approximate a normal distribution prior to inclusion in the regression models.

Next, we performed bivariate (Pearson) correlations to explore associations among continuous sociodemographic factors, health characteristics, and survey scores. The strength of correlations were interpreted using the following thresholds: negligible (0.0–0.3), low (0.3–0.5), moderate (0.5–0.7), high (0.7–0.9), and very high (0.9–1.0) [57].

To examine the moderating role of coping styles with diabetes distress, we conducted multiple-regression analyses focusing on two-way interactions with self-care behaviors. Only variables moderately or highly correlated with diabetes distress, coping styles, and/or self-care were included as covariates in the model. To ensure comparability across measures, all survey scores were standardized using z-transformation (mean = 0, standard deviation (SD)=1) before inclusion in the regression models.

For the moderation analysis, we centered the independent variable (type 1 and type 2 diabetes distress scores) and moderators (problem-based coping, emotion-based coping) and created an interaction term between the centered variables. We chose to center variables to reduce micro-multicollinearity (i.e., high correlation between the independent variable and moderator) and enhance interpretability of significant predictors [58]. In the model, we created a separate interaction term for diabetes distress and problem-based coping and another interaction term for diabetes distress and emotion-based coping. The interaction terms were added in step 2 of the final regression models. For both models, diabetes self-care was the primary outcome (dependent) variable. We calculated effect sizes for the regression models using the following formula $F^2 = R^2/(1-R^2)$, with a small effect=≥.02, medium effect=≥.15, and large effect=≥.35. We defined statistical significance as a p-value <0.05. We conducted all analyses with SPSS statistical software version 29.0 (SPSS Inc.).

## Results

The final sample included 256 participants, 38.7% (n = 99) reported a diagnosis of type 1 diabetes and 61.3% (n = 157) reported a diagnosis of type 2 diabetes (see Table 1). The mean diabetes distress score ± standard deviation for participants with type 1 diabetes was 2.4 ± 1.0, with 57.6% reporting moderate to high diabetes distress scores (see S1 Table). For participants with type 2 diabetes, the mean diabetes distress score was 2.4 ± 1.0, with 54.8% reporting moderate to high diabetes distress scores.

The mean A1C for participants with type 1 diabetes was 61.7 ± 17.5 mmol/mol (range = 31.1–119.7 mmol/mol; see Table 1) or 7.8 ± 1.6% (range = 5.0%−13.10%) and the mean A1C for participants with type 2 diabetes was 55.2 ± 15.3 mmol/mol (range = 32.2 to 114.2 mmol/mol) or 7.2 ± 1.4% (range = 5.1% to 12.6%). The mean diabetes duration for type 1 diabetes was 14.4 ± 8.9 years, with a range of 2.0 to 41.0 years, the mean BMI was 26.7 ± 6.5 kg/m² (range = 18.7–57.8 kg/m²), and the distribution of diabetes medications was 94.9% (n = 94) insulin and 5.1% (n = 5) insulin and oral medication(s). For participants with type 2 diabetes, the mean diabetes duration was 10.6 ± 8.7 years (range = 1.0 to 49.0 years), the mean BMI was 34.0 ± 8.9 kg/m² (range = 17.9–83.4 kg/m²), and the distribution of diabetes medications included 45.9% (n = 72) on oral medication(s), 24.2% (n = 38) on insulin and oral medication(s), 14.6% (n = 23) on no medications, 8.9% (n = 14) on insulin, and 6.4% (n = 10) on other medications (e.g., injectables).

The mean age ± SD of participants with type 1 and type 2 diabetes was 27.6 ± 12.9 years, with a range of 18.0 to 72.0 years, and 52.0 ± 15.4 years, with a range of 19.0 to 83.0 years, respectively (see Table 1). For participants with type 1 diabetes, 67.7% (n = 67) self-identified as women, 91.9% (n = 91) White, 98.0% (n = 97) non-Hispanic/non-Latino, 53.1% (n = 52) current undergraduate students, and 74.0% (n = 71) privately insured. For participants with type 2 diabetes, 58.6% (n = 92) self-identified as women, 81.5% (n = 128) White, 98.1% (n = 154) non-Hispanic/non-Latino, 17.2% (n = 27) high school education, and 54.8% (n = 86) privately insured.

### Correlational findings

Type 1 diabetes distress had low to moderate correlations with A1C levels (0.372; see Table 2), self-care (−0.282), depressive symptoms (0.616), problem-based coping (0.259), emotion-based coping (0.473), and social support (−0.331).

**Table 1. Participant Demographic and Health Characteristics (Total n = 256; Type 1 Diabetes n = 99; Type 2 Diabetes n = 157).**

| Variables | Total n (%) | Type 1 Diabetes n (%) | Type 2 Diabetes n (%) |
|---|---|---|---|
| **Age** (mean ± SD; years)[a] | 42.6 ± 18.7 | 27.6 ± 12.9 | 52.0 ± 15.4 |
| **Gender** | | | |
| Women | 159 (62.1) | 67 (67.7) | 92 (58.6) |
| Men | 96 (37.5) | 32 (32.3) | 64 (58.6) |
| Other | 1 (0) | 0 (0) | 1 (0.6) |
| Prefer not to answer | 0 (0) | 0 (0) | 0 (0) |
| **Race** | | | |
| American Indian or Alaska Native | 3 (1.2) | 0 (0) | 3 (1.9) |
| Asian | 12 (4.7) | 2 (2.0) | 10 (6.4) |
| Black or African American | 13 (5.1) | 3 (3.0) | 10 (6.4) |
| Native Hawaiian or Pacific Islander | 1 (0.4) | 0 (0) | 1 (0.6) |
| Two or more races | 4 (1.6) | 1 (1.0) | 3 (1.9) |
| White | 219 (85.5) | 91 (91.9) | 128 (81.5) |
| Another race not listed | 4 (1.6) | 2 (2.0) | 2 (1.3) |
| **Ethnicity** | | | |
| Hispanic/Latino | 5 (2.0) | 2 (2.9) | 3 (1.9) |
| Non-Hispanic/Non-Latino | 256 (98.0) | 97 (98.0) | 154 (98.1) |
| **Education**[b] | | | |
| Elementary school only | 1 (0.4) | 1 (1.0) | 0 (0) |
| Some high school, but did not finish | 2 (0.8) | 0 (0) | 2 (1.3) |
| High school | 32 (12.5) | 5 (5.1) | 27 (17.2) |
| Current undergraduate | 60 (23.5) | 52 (53.1) | 8 (5.1) |
| Some college | 30 (11.8) | 8 (8.2) | 22 (14.0) |
| Two-year degree | 26 (10.2) | 5 (5.1) | 21 (13.4) |
| Four-year degree | 36 (14.1) | 11 (11.2) | 25 (15.9) |
| Current graduate student | 8 (3.1) | 5 (5.1) | 3 (1.9) |
| Some graduate work | 9 (3.5) | 2 (2.0) | 7 (4.5) |
| Master's degree | 24 (9.4) | 6 (6.1) | 18 (11.5) |
| Doctoral/professional degree | 27 (10.5) | 3 (3.0) | 24 (15.3) |
| **Health Insurance Coverage**[c] | | | |
| No coverage | 12 (4.7) | 3 (3.0) | 9 (5.7) |
| Medicaid and/or Medicare | 63 (24.9) | 16 (16.7) | 47 (29.9) |
| Private insurance | 157 (62.1) | 71 (74.0) | 86 (54.8) |
| Other | 21 (8.3) | 6 (6.3) | 15 (9.6) |
| **Hemoglobin A$_{1c}$** | | | |
| Mean ± standard deviation (mmol/mol) | 58.5 ± 16.4 | 61.7 ± 17.5 | 55.2 ± 15.3 |
| Mean ± standard deviation (%) | 7.5 ± 1.5 | 7.8 ± 1.6 | 7.2 ± 1.4 |
| **Body Mass Index** (BMI; kg/m$^2$)[d] | 31.2 ± 8.8 | 26.7 ± 6.5 | 34.0 ± 8.9 |
| **Diabetes Duration** (Mean ± SD; years)[e] | 12.1 ± 8.9 | 14.4 ± 8.9 | 10.6 ± 8.7 |
| **Diabetes Medication** | | | |
| No medication(s) | 23 (9.0) | 0 (0) | 23 (14.6) |
| Oral medication(s) | 72 (28.1) | 0 (0) | 72 (45.9) |
| Insulin | 108 (42.2) | 94 (94.9) | 14 (8.9) |
| Insulin and oral medication(s) | 43 (16.8) | 5 (5.1) | 38 (24.2) |
| Other | 10 (3.9) | 0 (0) | 10 (6.4) |

Values missing for Age (n = 1)[a], Education (n = 1)[b], Health Insurance (n = 3)[c], BMI (n = 8)[d], Duration (n = 6)[e]; SD = Standard Deviation

**Table 2. Correlations between Health and Psychosocial Variables in Participants with Type 1 Diabetes (n = 99).**

| Variables | Age | Diabetes Duration | A1C | Self-Care | Depressive Symptoms | Diabetes Distress | Problem-Based Coping | Emotion-Based Coping | Diabetes Self-Efficacy |
|---|---|---|---|---|---|---|---|---|---|
| Diabetes Duration | 0.543*** | | | | | | | | |
| A1C | −0.063 | 0.006 | | | | | | | |
| Self-Care | 0.023 | −0.034 | −0.172 | | | | | | |
| Depressive Symptoms | −0.009 | −0.156 | 0.348*** | −0.200* | | | | | |
| Diabetes Distress | −0.144 | −0.147 | 0.372*** | −0.282** | 0.616*** | | | | |
| Problem-Based Coping | −0.016 | −0.116 | 0.081 | −0.020 | 0.246* | 0.259** | | | |
| Emotion-Based Coping | −0.159 | −0.139 | 0.171 | −0.292** | 0.484*** | 0.473*** | 0.143 | | |
| Diabetes Self-Efficacy | −0.093 | 0.103 | 0.004 | 0.245* | −0.244* | −0.039 | −0.181 | −0.018 | |
| Social Support | 0.032 | 0.022 | −0.173 | 0.327*** | −0.275** | −0.331*** | 0.027 | −0.129 | 0.354*** |

*$p < 0.05$, **$p < 0.01$, ***$p < 0.001$

Problem-based coping had low to moderate correlations with depressive symptoms (0.246) and emotion-based coping had low to moderate correlations with self-care (−0.292) and depressive symptoms (0.484). Variables with a correlation greater than (+) or (-).200 were included in the regression model.

Type 2 diabetes distress had low to moderate correlations with age (−0.278; see Table 3), A1C levels (0.158), self-care (−0.279), depressive symptoms (0.537), emotion-based coping (0.434), and social support (−0.395). Problem-based coping had low to moderate correlations with emotion-based coping (0.184). Emotion-based coping had low to moderate correlations with age (−0.181), self-care (−0.206), and depressive symptoms (0.494). Variables with a correlation greater than (+) or (-).200 were included in the regression model.

## Regression model findings

The overall type 1 diabetes regression model was significant (F(6, 92) = 3.444, p = 0.004). In Step 1 of the model, emotion-based coping (B = −0.233, p = 0.040, see Table 4) and social support (B = 0.268, p = 0.010) were independently associated with diabetes self-care; problem-based coping did not contribute to the overall model (B = 0.022, p = 0.823).

In Step 2 of the model, social support (B = 0.223, p = 0.026, see Table 4) remained a significant predictor, and the two-way interaction between diabetes distress and problem-based coping (B = 0.611, p = 0.011, see S2 Fig) along with

**Table 3. Correlations between Health and Psychosocial Variables in Participants with Type 2 Diabetes (n = 157).**

| Variables | Age | Diabetes Duration | A1C | Self-Care | Depressive Symptoms | Diabetes Distress | Problem-Based Coping | Emotion-Based Coping | Diabetes Self-Efficacy |
|---|---|---|---|---|---|---|---|---|---|
| Diabetes Duration | 0.395*** | | | | | | | | |
| A1C | 0.204** | 0.259** | | | | | | | |
| Self-Care | 0.213** | 0.091 | −0.060 | | | | | | |
| Depressive Symptoms | −0.093 | 0.059 | 0.351*** | −0.181* | | | | | |
| Diabetes Distress | −0.278*** | −0.099 | 0.158* | −0.279*** | 0.537*** | | | | |
| Problem-Based Coping | 0.040 | 0.031 | 0.148 | 0.107 | 0.111 | 0.110 | | | |
| Emotion-Based Coping | −0.181* | −0.121 | 0.092 | −0.206** | 0.494*** | 0.434*** | 0.184* | | |
| Diabetes Self-Efficacy | −0.031 | −0.060 | −0.134 | 0.233** | −0.264*** | −0.150 | −0.095 | −0.146 | |
| Social Support | 0.105 | 0.029 | −0.116 | 0.295*** | −0.426 | −0.395*** | −0.119 | −0.356*** | 0.257*** |

*$p < 0.05$, **$p < 0.01$, ***$p < 0.001$

**Table 4. Summary of Hierarchical regression analyses examining the moderating role of coping styles in the relationship among diabetes distress and self-care in participants with type 1 diabetes (n = 99).**

| Independent Variables | Step 1 | | | | | | Step 2 | | | | | |
|---|---|---|---|---|---|---|---|---|---|---|---|---|
| | B | SE | Beta | p | R² | F² | B | SE | Beta | p | R² | F² |
| Hemoglobin A1c | −0.891 | 1.247 | −0.074 | 0.476 | 0.130 | 0.149 | −1.255 | 1.191 | −0.104 | 0.295 | 0.225 | 0.290 |
| Depressive Symptoms | 0.071 | 0.128 | 0.071 | 0.579 | | | −0.009 | 0.127 | −0.009 | 0.946 | | |
| Diabetes Distress | −0.106 | 0.131 | −0.106 | 0.423 | | | −0.142 | 0.127 | −0.142 | 0.268 | | |
| Problem-Based Coping | 0.022 | 0.099 | 0.022 | 0.823 | | | 0.145 | 0.099 | 0.145 | 0.149 | | |
| Emotion-Based Coping | −0.233 | 0.112 | −0.233 | 0.040 | | | −0.184 | 0.106 | −0.184 | 0.087 | | |
| Social Support | 0.268 | 0.102 | 0.268 | 0.010 | | | 0.223 | 0.099 | 0.223 | 0.026 | | |
| Diabetes Distress x Problem-Based Coping | | | | | | | 0.611 | 0.237 | 0.244 | 0.011 | | |
| Diabetes Distress x Emotion-Based Coping | | | | | | | 0.379 | 0.156 | 0.244 | 0.017 | | |

B=Unstandardized Coefficient, SE=Standard Error, Beta=Standardized Coefficient

the two-way interaction between diabetes distress and emotion-based coping (B = 0.379, p = 0.017, see S3 Fig) were significant (R2Δ = 0.105, p = 0.002). These two-way interactions suggest problem-based coping mitigated or buffered the negative effect of diabetes distress on self-care while emotion-based coping amplified or exacerbated the negative effect of diabetes distress on diabetes self-care. Overall, this model had an effect size of $F^2 = 0.290$, indicating a medium effect.

The overall type 2 diabetes regression model was significant (F(7, 149) = 3.444, p < 0.001). In Step 1 of the model, problem-based coping (B = 0.166, p = 0.033, see Table 5) and social support (B = 0.226, p = 0.009) were independently associated with diabetes self-care; emotion-based coping did not contribute to the overall model (B = −0.087, p = 0.340).

In Step 2 of the model, problem-based coping (B = 0.198, p = 0.010) and social support (B = 0.189, p = 0.029, see Table 5) remained significant predictors, and the two-way interaction between diabetes distress and problem-based coping (B = 0.388, p = 0.012, see S4 Fig) was significant (R2Δ = 0.044, p = 0.018). This two-way interaction suggests that problem-based coping buffered the negative effect of diabetes distress on self-care. Overall, this model had an effect size of $F^2 = 0.198$, indicating a medium effect.

**Table 5. Summary of hierarchical regression analyses examining the moderating role of coping styles in the relationship among diabetes distress and self-care in participants with type 2 diabetes (n = 157).**

| Independent Variables | Step 1 | | | | | | Step 2 | | | | | |
|---|---|---|---|---|---|---|---|---|---|---|---|---|
| | B | SE | Beta | p | R² | F² | B | SE | Beta | p | R² | F² |
| Age | 0.009 | 0.005 | 0.147 | 0.072 | 0.130 | 0.149 | 0.009 | 0.005 | 0.142 | 0.076 | 0.165 | 0.198 |
| Hemoglobin A1c | −1.005 | 1.080 | −0.078 | 0.353 | | | −0.921 | 1.060 | −0.071 | 0.386 | | |
| Depressive Symptoms | 0.061 | 0.102 | 0.061 | 0.549 | | | 0.026 | 0.101 | 0.026 | 0.799 | | |
| Diabetes Distress | −0.150 | 0.096 | −0.150 | 0.118 | | | −0.178 | 0.095 | −0.178 | 0.062 | | |
| Problem-Based Coping | 0.166 | 0.077 | 0.166 | 0.033 | | | 0.198 | 0.076 | 0.198 | 0.010 | | |
| Emotion-Based Coping | −0.087 | 0.091 | −0.087 | 0.340 | | | −0.076 | 0.095 | −0.076 | 0.397 | | |
| Social Support | 0.226 | 0.086 | 0.226 | 0.009 | | | 0.189 | 0.086 | 0.189 | 0.029 | | |
| Diabetes Distress x Problem-Based Coping | | | | | | | 0.388 | 0.153 | 0.194 | 0.012 | | |
| Diabetes Distress x Emotion-Based Coping | | | | | | | 0.094 | 0.112 | 0.065 | 0.404 | | |

B=Unstandardized Coefficient, SE=Standard Error, Beta=Standardized Coefficient

## Discussion

This study examined the moderating role of coping styles in the relationship between diabetes distress and self-care in adults with type 1 and type 2 diabetes. Consistent with our first hypothesis, we found that higher levels of problem-based coping moderated the negative effect of diabetes distress on self-care behaviors for both participants with type 1 and type 2 diabetes. However, we only found evidence supporting our second hypothesis that higher levels of emotion-based coping exacerbated the negative effect of diabetes distress on self-care behaviors in participants with type 1 diabetes. These findings indicate problem-based coping strategies may buffer the negative effects of diabetes distress on self-care behaviors in adults with type 1 and type 2 diabetes living in Appalachian Ohio. For adults with type 1 diabetes living here, emotion-based coping strategies may amplify the negative effects of diabetes distress on self-care behaviors.

These findings align with prior research on coping strategies and diabetes distress [33,59,60]. Across multiple studies, greater use of problem-based coping strategies was associated with lower diabetes distress and more self-care behaviors [33,59,60]. In contrast, greater use of emotion-based coping strategies was associated with higher diabetes distress and fewer self-care behaviors [33,59,60]. Altogether, these findings reinforce the importance of interventions designed to enhance coping strategies to reduce diabetes distress and improve self-management.

Diabetes self-management education and support (DSMES) is generally considered the first-line therapy for people with moderate to high levels of diabetes distress [61,62]. DSMES is evidence-based and adaptable to all care models including independent practice, community-based care, large group practices, technology-enabled care models, and more [62–64]. People with diabetes should receive DSMES at diagnosis, annually and/or when not meeting treatment goals, when complicating factors develop (e.g., medical, functional, and psychosocial), and when transitions in life and care occur. For people with moderate to high diabetes distress, DSMES can target specific aspects of diabetes self-care that contribute to frustrations, worries, and concerns, such as glucose monitoring, meal planning, hypoglycemia [62]. An important core content area in DSMES is problem-solving, which is a skill aimed at helping people with diabetes identify their barriers to self-care, develop practical solutions, and adapt to the everchanging needs of a complex, progressive condition [65]. Problem-solving in DSMES is similar to problem-based coping skills, such that they both focus on problem identification, a structured approach to address the problem, personal agency and health behavior change, and goal-driven outcomes. Based on our findings and prior research, which suggest that problem-based coping buffers the negative effect of diabetes distress on self-care, incorporating problem-based coping skills into DSMES through problem-solving therapy could reduce diabetes distress. Supporting this, a 2017 pilot randomized controlled trial with 40 participants with diabetic retinopathy and diabetes distress demonstrated that problem-solving therapy techniques reduced regimen-related diabetes distress, depressive symptoms, and A1C levels compared to the control group [66]. Thus, integrating problem-based coping techniques in DSMES, including strategies like information seeking, setting priorities, problem-solving, time management, assertive communication, and boundary setting, may improve outcomes related to diabetes distress.

If DSMES does not reduce moderate to high levels of diabetes distress, people with diabetes should be referred to qualified behavioral health professionals for evaluation and treatment [61]. Behavioral health is recommended as the second-line therapy for moderate to high diabetes distress. A 2024 meta-analysis of 16 psychological interventions (n = 1639) on diabetes distress demonstrated a medium reduction in diabetes distress compared to control conditions (Standard Mean Difference = −0.56; 95% CI = −0.90,-0.22; P = 0.001); however, the studies showed high heterogeneity ($I^2$ = 91%, $p < 0.001$) [67]. The evidence-based psychological interventions in the meta-analysis included cognitive behavioral therapy (CBT) [68], mindfulness-based therapy [69–73], acceptance and commitment therapy [74,75], motivational interviewing [76], and problem solving therapy [66,77]. Subgroup analyses showed intervention effects were bolstered by including a group format, a technology component, and psychologists delivering the intervention [67]. Another randomized controlled trial specifically for adults with type 1 diabetes compared an emotion-focused intervention to an educational intervention, and found both approaches significantly reduced diabetes distress [78,79]. Additional analyses found increasing awareness of emotions associated with diabetes and improving diabetes-related problem-solving skills decreased diabetes distress [79].

## Limitations

This study has several limitations. First, the study sample lacked racial and ethnic diversity; although, the racial and ethnic distribution is representative of southeastern Ohio (93% White). The lack of diversity may limit the generalizability of the findings to other subgroups of the population based on race, ethnicity, and geographic location. Second, participants completed surveys either through an online platform or mailed packets, potentially limiting participation among individuals with lower literacy levels. Third, this study relied on self-reported health data, which is subject to recall bias and social desirability bias. Participants also volunteered to participate in the study, which increased the risk for self-selection bias because individuals who participated in the study may have differed from those who did not. We recruited participants from the University, which is the largest employer in the region; however, this recruitment source likely increased the percentage of current undergraduate students and participants with a graduate or professional degree than would be expected in the general population. Finally, the cross-sectional study design limits the ability to determine temporal associations or infer causality suggested by the moderation analysis. While the proposed causal pathway is plausible, future longitudinal research with larger, more diverse samples are required to confirm how coping styles moderate the relationship between diabetes distress and self-care.

## Conclusion

In conclusion, this study demonstrates the role for problem-based coping in buffering the negative effects of diabetes distress on self-care behaviors in adults with type 1 and type 2 diabetes living in Appalachian Ohio. Furthermore, emotion-based coping amplified the negative effect of diabetes distress on self-care behaviors, but only in participants with type 1 diabetes. These findings underscore the importance of addressing psychosocial factors in diabetes management. Moreover, these findings suggest that evidence-based psychosocial interventions that have been effective in other regions could be adapted and implemented in rural Appalachian Ohio to reduce diabetes distress and improve self-care behaviors.

## Supporting information

**S1 Fig. Step-by-Step Flow Diagram of Diabetes Diagnosis Confirmation.**
(TIF)

**S2 Fig. Interaction of problem-based coping and diabetes distress predicting diabetes self-care in adults with type 1 diabetes (n = 99).** Legend: Lines represent predicted self-care scores at low, mean, and high levels of problem-based coping and diabetes distress (values represent ±1 SD around the mean). Problem-Based Coping (1 = Low, 2 = Mean, 3 = High).
(TIF)

**S3 Fig. Interaction of emotion-based coping and diabetes distress predicting diabetes self-care in adults with type 1 diabetes (n = 99).** Legend: Lines represent predicted self-care scores at low, mean, and high levels of emotion-based coping and diabetes distress (values represent ±1 SD around the mean). Emotion-Based Coping (1 = Low, 2 = Mean, 3 = High).
(TIF)

**S4 Fig. Interaction of problem-based coping and diabetes distress predicting diabetes self-care in adults with type 2 diabetes (n = 157).** Legend: Lines represent predicted self-care scores at low, mean, and high levels of problem-based coping and diabetes distress (values represent ±1 SD around the mean). Problem-Based Coping (1 = Low, 2 = Mean, 3 = High).
(TIF)

**S1 Table. Means and Standard Deviations of Psychosocial Measures among Participants with Type 1 and Type 2 Diabetes (n = 256).** Legend: Diabetes Distress Total Scores = Type 1 Diabetes Distress Scale and Subscales and Type 2 Diabetes Distress Scale and Subscales; Depressive Symptoms = Patient Health Questionnaire-9 (PHQ-9); Diabetes Self-Care = Self-Care Inventory-Revised; Problem-Based Coping and Emotion-Based Coping = Coping Styles; Diabetes Self-Efficacy = Confidence in Diabetes Self-Care Scale; Social Support = Medical Outcomes Study Social Support Survey. (DOCX)

## Author contributions

**Conceptualization:** Elizabeth Ann Beverly.

**Data curation:** Elizabeth Ann Beverly.

**Formal analysis:** Elizabeth Ann Beverly, Samuel Miller, Lauren Schmidtgesling, Kacey Whistler.

**Funding acquisition:** Elizabeth Ann Beverly.

**Investigation:** Elizabeth Ann Beverly, Samuel Miller, Lauren Schmidtgesling, Kacey Whistler.

**Methodology:** Elizabeth Ann Beverly, Samuel Miller, Lauren Schmidtgesling, Kacey Whistler.

**Project administration:** Elizabeth Ann Beverly, Samuel Miller, Lauren Schmidtgesling, Kacey Whistler.

**Resources:** Elizabeth Ann Beverly.

**Software:** Elizabeth Ann Beverly.

**Supervision:** Elizabeth Ann Beverly.

**Validation:** Elizabeth Ann Beverly.

**Visualization:** Elizabeth Ann Beverly.

**Writing – original draft:** Elizabeth Ann Beverly, Samuel Miller, Lauren Schmidtgesling, Kacey Whistler.

**Writing – review & editing:** Elizabeth Ann Beverly, Samuel Miller, Lauren Schmidtgesling, Kacey Whistler.

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
