## [Decision Letter · Decision Letter 0]

12 May 2025

PLOS ONE

Dear Dr. Beverly,

Thank you for submitting your manuscript to PLOS ONE. After careful consideration, we feel that it has merit but does not fully meet PLOS ONE’s publication criteria as it currently stands. Therefore, we invite you to submit a revised version of the manuscript that addresses the points raised during the review process.

Although the manuscript is well-written and addresses an important topic, the manuscript requires few minor edits as raised by the expert reviewers. Taking into account changes could undoubtedly enhance the manuscript's readability and quality.

We look forward to receiving your revised manuscript.

Kind regards,

Mukhtar Ansari, Ph D

Academic Editor

PLOS ONE

“American Osteopathic Association Grant: Osteopathic Approach to Diabetes: Pathophysiology, Treatment, Outcomes and Complication Grant (No. 1291708718)”

3. In the online submission form, you indicated that [All relevant data are within the manuscript and supporting tables. The dataset underlying the results presented in the study are available upon request from the corresponding author.].

Additional Editor Comments:

Thanks for submitting a well-written manuscript that covers a very relevant subject. However, the manuscript requires a few minor revisions as suggested by the expert reviewers before it can be taken into consideration. Taking into account changes could undoubtedly enhance the manuscript's readability and quality.

Reviewers' comments:

Reviewer's Responses to Questions

**Comments to the Author**

1. Is the manuscript technically sound, and do the data support the conclusions?

Reviewer #1: Yes

Reviewer #2: Partly

2. Has the statistical analysis been performed appropriately and rigorously?

Reviewer #1: Yes

Reviewer #2: No

3. Have the authors made all data underlying the findings in their manuscript fully available?

Reviewer #1: Yes

Reviewer #2: No

4. Is the manuscript presented in an intelligible fashion and written in standard English?

Reviewer #1: Yes

Reviewer #2: Yes

Reviewer #1: This manuscript addresses a highly relevant and timely topic: the interaction between coping styles and diabetes distress in relation to self-care behaviors among adults with type 1 and type 2 diabetes. The study design is appropriate, and the sample size is adequate. The use of validated tools enhances the credibility of the findings.

Strengths:

Clear and justified research questions and hypotheses.

Robust statistical methods with a clear rationale for interaction testing.

Important clinical implications for diabetes education and behavioral interventions.

Well-articulated discussion connecting findings to existing literature and practice.

Areas for Minor Improvement:

Clarify Missing Data: While the manuscript acknowledges some missing values in demographics and lab measures, it would help to elaborate briefly on how missing data were handled in the regression models.

Figure/Tables Enhancements: If space permits, consider including a simple interaction plot showing the moderating effect of problem-focused coping on the diabetes distress-self-care relationship.

Language/Typo: In the Abstract, line 32 – "This purpose of this study" should be corrected to "The purpose of this study".

Limitations: The limitations section is candid and well-written, though it may benefit from explicitly noting the potential for self-selection bias due to voluntary survey participation.

Data Transparency: Although the data availability is noted, the authors may consider adding a brief supplemental table with mean scores and standard deviations for each major construct (e.g., coping, self-care), to facilitate understanding and reproducibility.

Overall, this is a valuable contribution to the literature on psychosocial factors in diabetes management. With some minor revisions, it is suitable for publication.

Reviewer #2: Summary:

The study evaluates effects of degree of diabetes distress and coping strategies on diabetes self care, measured by questionnaires in a arbitrary selection of Ohio residents. The study is interesting, although novelty needs to be clarified. Furthermore, T1D and T2D should be separated as they are two distinctly different diseases.

Major:

1. The participant recruitment could be clarified. Where were participant recruited specifically and who were the advertisements targeting and where were flyers distributed, in specific communities or cities etc? What measures were taken to ascertain a sample representative of the overall Ohio population?

2. Please include a step-by-step flow chart of all steps taken to determine if a participant had or did not have a diabetes diagnose as supplementary material. Each step should include the n at that stage to illustrate how many are lost in the process. There is both a risk of defining the wrong individuals, but perhaps even more so, to define individuals with diabetes as non-diabetics due to insufficient reporting. This may affect the representativity of the study as participants with low insight may be missed.

3. Please separate individuals with T1D and T2D in this study (all data analysis). They are two completely different diseases, with different treatments, etiology and health risks. Furthermore, it is not appropriate to combine two different scoring systems into one (diabetes distress in T1D and T2D) as this will bias the regression analysis.

4. Please specify the primary outcome of the study in the statistics section. The title currently suggests that coping is an exposure and distress is the outcome. This makes sense but is not currently what is tested in the regression model (where distress and coping are both exposures.

5. As discussed there are multiple intervention studies targeting diabetes distress, but it is not clearly communicated to me what new information this cross-sectional study contributes with in relation to what has already been demonstrated in human interventional experiments for T1D and T2D respectively. While it is suggested on line 337 what direction to take going forward, it appears to me that several studies already discussed have already successfully done what is suggested by the authors? Perhaps this study is more a confirmatory study, indicating that previous successful intervention studies may also hold feasibility for residents in Ohio?

Minor:

1. Are “indian” and “race” adequate terminology? Consider eg. native Americans and ethnicity as alternative terms.

2. 9.5% of participants have a doctoral degree, is this representative of residens of Ohio? Please discuss.

3. Approx 1/4 with T2D appear to be on insulin, is this representative of the Ohio population, it appears to be a high proportion in relation to current guidelines. What does this say about sample selection? Please discuss.

4. Clarify gender and occupation in table 2 (what does low/high mean). Is duration diabetes duration?

5. Please elaborate on the coping style questionnaire, what is the outcome? Are participants separated in either type or do you ge a final score of up to 60 points (15*4), where low is emotional and high is pragmatic?

6. What is the cutoffs for moderate and high diabets distress, add to methods with reference.

7. Please clarify how you handled non-normally distributed variables as A1c. The outliers around >100 mmol/mol will likely pivot alla analysis and a sensitivity analysis should exclude these individuals to validate current associations found for A1c.

8. Consider if z-transformation of reported scales may be more appropriate to include in linear regression models than untransformed scales.

**Do you want your identity to be public for this peer review?** For information about this choice, including consent withdrawal, please see our Privacy Policy

Reviewer #1: **Yes: ** Elabbass Ali Abdelmahmuod

Reviewer #2: No

---

## [Author Response · Author response to Decision Letter 1]

5 Nov 2025

Response: Thank you for sharing the style templates. We have reviewed the formatting and made revisions to adhere to the style requirements and file naming.

“American Osteopathic Association Grant: Osteopathic Approach to Diabetes: Pathophysiology, Treatment, Outcomes and Complication Grant (No. 1291708718)”

Response: Thank you. This is correct. The funder had no role in the study design, data collection and analysis, decision to publish, or preparation of the manuscript. We have included this in the cover letter.

3. In the online submission form, you indicated that [All relevant data are within the manuscript and supporting tables. The dataset underlying the results presented in the study are available upon request from the corresponding author.].

Response: We have uploaded all relevant data to a public repository. The dataset is in Mendeley. Here in the information for the public repository: Beverly, Elizabeth (2025), “EBeverlyDiabetesDistressCopingDataset”, Mendeley Data, V1, doi: 10.17632/rrx6stxcfk.1 or https://data.mendeley.com/datasets/rrx6stxcfk/1

Response: Thank you for this comment. We have uploaded all relevant data to a public repository. The dataset is in Mendeley. Here in the information for the public repository: Beverly, Elizabeth (2025), “EBeverlyDiabetesDistressCopingDataset”, Mendeley Data, V1, doi: 10.17632/rrx6stxcfk.1 or https://data.mendeley.com/datasets/rrx6stxcfk/1

Response: We have looked up every article on our reference list to see if any articles were retracted. Our reference list is complete and correct. None of our cited articles have been retracted. We have two articles listed with Erratums (Young-Hyman D et al., 2016; Ryan D et al., 2020). Additionally, we have three articles that are not indexed in PubMed (Ngan et al., 2023; Pearson et al., 2018; Schroevers et al., 2015). We included those articles because they were mindfulness based randomized controlled trials and an acceptance-based diabetes education randomized controlled trial, which are pertinent to diabetes distress.

Additional Editor Comments:

Thanks for submitting a well-written manuscript that covers a very relevant subject. However, the manuscript requires a few minor revisions as suggested by the expert reviewers before it can be taken into consideration. Taking into account changes could undoubtedly enhance the manuscript's readability and quality.

Reviewers' comments:

Reviewer's Responses to Questions

Comments to the Author

1. Is the manuscript technically sound, and do the data support the conclusions?

Reviewer #1: Yes

Reviewer #2: Partly

2. Has the statistical analysis been performed appropriately and rigorously?

Reviewer #1: Yes

Reviewer #2: No

Response: We have revised all statistical analyses. We believe they have been performed appropriately and rigorously in this revision.

3. Have the authors made all data underlying the findings in their manuscript fully available?

Reviewer #1: Yes

Reviewer #2: No

Response: We have uploaded all relevant data to a public repository. The dataset is in Mendeley. Here in the information for the public repository: Beverly, Elizabeth (2025), “EBeverlyDiabetesDistressCopingDataset”, Mendeley Data, V1, doi: 10.17632/rrx6stxcfk.1 or https://data.mendeley.com/datasets/rrx6stxcfk/1

4. Is the manuscript presented in an intelligible fashion and written in standard English?

Reviewer #1: Yes

Reviewer #2: Yes

5. Review Comments to the Author

Reviewer #1: This manuscript addresses a highly relevant and timely topic: the interaction between coping styles and diabetes distress in relation to self-care behaviors among adults with type 1 and type 2 diabetes. The study design is appropriate, and the sample size is adequate. The use of validated tools enhances the credibility of the findings.

Response: Thank you for the positive feedback.

Strengths:

Clear and justified research questions and hypotheses.

Robust statistical methods with a clear rationale for interaction testing.

Important clinical implications for diabetes education and behavioral interventions.

Well-articulated discussion connecting findings to existing literature and practice.

Response: Thank you for the positive feedback.

Areas for Minor Improvement:

Clarify Missing Data: While the manuscript acknowledges some missing values in demographics and lab measures, it would help to elaborate briefly on how missing data were handled in the regression models.

Response: Thank you for asking this question. For this revision, we were asked to conduct a multi-step data verification process to ensure accurate classification of participants with type 1 and type 2 diabetes. Our process included sequential checks for medication congruence, scale completion, consistency between age and diabetes duration, plausible A1C values, and duplicate entries prior to final analyses (see Supplemental Figure). As a result, we excluded 69 cases from our previous sample size (n=31 type 1 diabetes, n=38 with type 2 diabetes). The removal of these cases also removed the majority of cases with missing data. In our regression models, the only variable with missing data is age (n=1).

Figure/Tables Enhancements: If space permits, consider including a simple interaction plot showing the moderating effect of problem-focused coping on the diabetes distress-self-care relationship.

Response: Thank you for this comment. We have included three interaction plots as supplemental information to show the moderating effect of problem-based coping and emotion-based coping on the diabetes distress-self-care relationship.

Language/Typo: In the Abstract, line 32 – "This purpose of this study" should be corrected to "The purpose of this study".

Response: Thank you for noting this typo. We have corrected it in the Abstract.

Limitations: The limitations section is candid and well-written, though it may benefit from explicitly noting the potential for self-selection bias due to voluntary survey participation.

Response: Thank you. We have added self-selection as a limitation to our study.

Data Transparency: Although the data availability is noted, the authors may consider adding a brief supplemental table with mean scores and standard deviations for each major construct (e.g., coping, self-care), to facilitate understanding and reproducibility.

Response: Excellent suggestion. We have added this table as supplemental information. In addition, we have uploaded our dataset to a publicly available registry.

Overall, this is a valuable contribution to the literature on psychosocial factors in diabetes management. With some minor revisions, it is suitable for publication.

Response: Thank you for the kind feedback.

Reviewer #2: Summary:

The study evaluates effects of degree of diabetes distress and coping strategies on diabetes self-care, measured by questionnaires in an arbitrary selection of Ohio residents. The study is interesting, although novelty needs to be clarified. Furthermore, T1D and T2D should be separated as they are two distinctly different diseases.

Response: Thank you for this comment. We have revised our analyses and conducted separate models for type 1 and type 2 diabetes. In addition, we clarified that this study was a confirmation study in a unique population of Appalachian Ohio.

Major:

1. The participant recruitment could be clarified. Where were participants recruited specifically and who were the advertisements targeting and where were flyers distributed, in specific communities or cities etc? What measures were taken to ascertain a sample representative of the overall Ohio population?

Response: Thank you for this comment. We have included additional information about recruitment. This study focuses on southeastern Ohio, which is a seven-county region in southeastern Appalachian Ohio: “Recruitment emails were distributed to all employees at the University, which is the largest employer in the seven-county region, and to individuals listed in the Diabetes Institute registry who had previously given permission to be contacted for future research opportunities. In addition, study flyers were posted in public locations, including grocery stores, libraries, primary care offices, and the seven country health departments. Finally, written advertisements were printed in the local newspaper and broadcast on the local public radio station to increase recruitment reach and participation across the seven-county region.”

2. Please include a step-by-step flow chart of all steps taken to determine if a participant had or did not have a diabetes diagnosis as supplementary material. Each step should include the n at that stage to illustrate how many are lost in the process. There is both a risk of defining the wrong individuals, but perhaps even more so, to define individuals with diabetes as non-diabetics due to insufficient reporting. This may affect the representativity of the study as participants with low insight may be missed.

Response: Thank you for this recommendation. We conducted a multi-step data verification process to ensure accurate classification of participants with type 1 and type 2 diabetes. Our process included sequential checks for medication congruence, scale completion, consistency between age and diabetes duration, plausible A1C values, and duplicate entries prior to final analyses (see Supplemental Figure). As a result, we excluded 69 cases from our previous sample size (n=31 type 1 diabetes, n=38 with type 2 diabetes). The removal of these cases also removed most cases with missing data. We believe this multi-step process determined whether or not a person had diabetes and which type of diabetes.

3. Please separate individuals with T1D and T2D in this study (all data analysis). They are two completely different diseases, with different treatments, etiology and health risks. Furthermore, it is not appropriate to combine two different scoring systems into one (diabetes distress in T1D and T2D) as this will bias the regression analysis.

Response: Thank you for this comment. We have separated the analyses and the models. All data are presented separately.

4. Please specify the primary outcome of the study in the statistics section. The title currently suggests that coping is an exposure and distress is the outcome. This makes sense but is not currently what is tested in the regression model (where distress and coping are both exposures.

Response: We have clarified the primary outcome in the statistics section as well as in the title. Thank you.

5. As discussed there are multiple intervention studies targeting diabetes distress, but it is not clearly communicated to me what new information this cross-sectional study contributes with in relation to what has already been demonstrated in human interventional experiments for T1D and T2D respectively. While it is suggested on line 337 what direction to take going forward, it appears to me that several studies already discussed have already successfully done what is suggested by the authors? Perhaps this study is more a confirmatory study, indicating that previous successful intervention studies may also hold feasibility for residents in Ohio?

Response: We agree that this is not a new finding, and previous intervention studies have shown that addressing coping strategies can reduce diabetes distress. The purpose of this paper was to replicate and confirm prior research and contribute to the literature on this topic. Furthermore, we aimed to replicate a

---

## [Editor Report · Decision Letter 1]

10 Nov 2025

How Coping Styles Moderate the Relationship between Diabetes Distress and Self-Care in Adults with Diabetes in Appalachian Ohio: A Cross-Sectional Survey Study

PONE-D-25-08613R1

Dear Dr. Elizabeth Ann Beverly,

We’re pleased to inform you that your manuscript has been judged scientifically suitable for publication and will be formally accepted for publication once it meets all outstanding technical requirements.

Kind regards,

Mukhtar Ansari, Ph D

Academic Editor

PLOS ONE

Additional Editor Comments (optional):

I would like to thank the authors for your diligent effort in addressing the comments provided by the expert reviewers.
---

## [Editor Report · Acceptance letter]

PONE-D-25-08613R1

PLOS ONE

Dear Dr. Beverly,

I'm pleased to inform you that your manuscript has been deemed suitable for publication in PLOS ONE. Congratulations! Your manuscript is now being handed over to our production team.

Kind regards,

on behalf of

Dr Mukhtar Ansari

Academic Editor

PLOS ONE